# Working with what you have: How the East Africa Preterm Birth Initiative used gestational age data from facility maternity registers

Lara Miller[1]*, Phillip Wanduru[2], Nicole Santos[1], Elizabeth Butrick[1], Peter Waiswa[2], Phelgona Otieno[3], Dilys Walker[1,4]

1 Institute for Global Health Sciences, University of California, San Francisco, San Francisco, California, United States of America, 2 School of Public Health, Makerere University, Kampala, Uganda, 3 Kenya Medical Research Institute, Nairobi, Kenya, 4 Department of Obstetrics and Gynecology, University of California, San Francisco, San Francisco, California, United States of America

* lara.miller@ucsf.edu

**Data Availability Statement:** The dataset is available in the Dryad Digital Repository https://doi.org/10.7272/Q6833Q63.

## Abstract

### Objective

Preterm birth is the primary driver of neonatal mortality worldwide, but it is defined by gestational age (GA) which is challenging to accurately assess in low-resource settings. In a commitment to reducing preterm birth while reinforcing and strengthening facility data sources, the East Africa Preterm Birth Initiative (PTBi-EA) chose eligibility criteria that combined GA and birth weight. This analysis evaluated the quality of the GA data as recorded in maternity registers in PTBi-EA study facilities and the strength of the PTBi-EA eligibility criteria.

### Methods

We conducted a retrospective analysis of maternity register data from March–September 2016. GA data from 23 study facilities in Migori, Kenya and the Busoga Region of Uganda were evaluated for completeness (variable present), consistency (recorded versus calculated GA), and plausibility (falling within the 3rd and 97th birth weight percentiles for GA of the INTERGROWTH-21st Newborn Birth Weight Standards). Preterm birth rates were calculated using: 1) recorded GA <37 weeks, 2) recorded GA <37 weeks, excluding implausible GAs, 3) birth weight <2500g, and 4) PTBi-EA eligibility criteria of <2500g and between 2500g and 3000g if the recorded GA is <37 weeks.

### Results

In both countries, GA was the least recorded variable in the maternity register (77.6%). Recorded and calculated GA (Kenya only) were consistent in 29.5% of births. Implausible GAs accounted for 11.7% of births. The four preterm birth rates were 1) 14.5%, 2) 10.6%, 3) 9.6%, 4) 13.4%.

**Funding:** The East Africa Preterm Birth Initiative was generously funded by the Bill and Melinda Gates Foundation.

**Competing interests:** The authors have declared that no competing interests exist.

## Conclusions

Maternity register GA data presented quality concerns in PTBi-EA study sites. The PTBi-EA eligibility criteria of <2500g and between 2500g and 3000g if the recorded GA is <37 weeks accommodated these concerns by using both birth weight and GA, balancing issues of accuracy and completeness with practical applicability.

## Introduction

Defined by the World Health Organization (WHO) as a baby born before 37 weeks gestational age (GA), preterm birth is the leading cause of neonatal mortality in Kenya and Uganda as well as worldwide [1–3]. Preterm birth rates rely on GA measurements assessed and recorded by frontline health workers and methods of GA assessment vary widely in sensitivity, specificity, and practical applicability. A first trimester ultrasound where fetal crown-rump length is measured is the international gold standard, but requires a significant upfront capital investment and specially trained technicians, making it cost prohibitive in many low-resource settings [4, 5]. Early ultrasound is also contingent upon women seeking antenatal care (ANC) in the first trimester of pregnancy, which only 19% of women do in Kenya and 29% in Uganda with an average first visit of 5.4 and 4.7 months, respectively [6, 7].

Non-ultrasound GA measurements are predominantly used in low-resource settings, but with myriad barriers to an accurate assessment. Fundal height measurements require a tape measurer, have a margin of error of +/- 3 weeks, and rely on the mother bringing her ANC documentation to her delivery, resulting in high levels of missing data [8, 9]. Post-partum infant examinations, such as Ballard or Dubowitz scores, require a skilled examiner and have been shown to skew towards overestimating GA and therefore underestimating preterm birth [10–14]. Using last menstrual period (LMP) to calculate GA based on the day of delivery is the most frequently used method worldwide, but it is often inaccurate due to patient recall error, imprecise due to variations in menstrual cycles, and prone to calculations errors [15–17]. Studies have shown that when LMP GAs are compared to ultrasound GAs, they have been inaccurate in up to half of all births [13, 18, 19]. This is an especially poor method of assessing GA in the intrapartum period due to the increase of recall bias proportional to the progression of pregnancy [20].

Studies conducted in low-resource settings generally conclude that there is no substitute for a first-trimester ultrasound in achieving an accurate GA, yet the impracticality of universal ultrasound for all pregnant women leaves healthcare workers with inadequate clinical data and preterm birth researchers with an unclear approach to eligibility criteria.

In order to reduce neonatal mortality rates from 19.9 and 19.6 per 1,000 live births, in Kenya and Uganda respectively, to the Sustainable Development Goal of 12 per 1,000 live births by 2030, both countries have prioritized intrapartum care of mothers in preterm labor and postpartum care of preterm infants [21, 22]. The East Africa Preterm Birth Initiative (PTBi-EA) was part of that effort and implemented an intrapartum package of interventions in Migori, Kenya and the Busoga Region of Uganda to improve the quality of preterm care and increase survival of preterm neonates.

As an implementation science study, PTBi-EA was dedicated to strengthening and using routinely collected data sources rather than implementing a protocol of early ultrasound for all study participants [23, 24]. The study therefore relied on the GA recorded by the healthcare

workers in the maternity register, a large handwritten ledger where demographic and out-comes data are recorded for every patient admitted to the maternity ward within each facility.

This nested analysis looked at the GA, birth weight, sex, and birth outcomes data from the maternity registers of the 23 PTBi-EA study facilities to determine the quality of the GA data and its reliability as a source for categorizing babies as term or preterm. We also evaluated various approaches to preterm birth eligibility criteria and calculating preterm birth rates given GA data limitations.

## Methods

### Overview of PTBi-EA study

In a collaboration between the University of California, San Francisco (UCSF), the Kenya Medical Research Institute, and Makerere University in Uganda, PTBi-EA implemented a package of intrapartum and immediate postpartum interventions aimed at improving the quality of maternity and newborn care. The study was a cluster randomized control trial (CRCT) targeting healthcare workers in the maternity and newborn units of 10 intervention facilities, 10 control facilities, and 3 referral facilities that received the intervention but were not included in the primary analysis. The study facilities were mostly public, government hospitals and healthcare centers, staffed predominantly by nurse-midwives, nurses, and clinical officers. Success was measured by the comparison of fresh stillbirth and neonatal mortality among preterm babies born at the intervention versus control facilities. Results of the CRCT saw a 34% decreased odds in neonatal mortality in the intervention sites among eligible infants and are published elsewhere [25].

The intervention package included data strengthening, introduction of a modified version of the WHO Safe Childbirth Checklist (mSCC), a quality improvement (QI) collaborative, and PRONTO simulation and team training. To address data quality concerns, all facilities received the mSCC and on-going data strengthening support which included a 2-day training during the baseline data collection period in an effort to improve baseline data for more accurate comparisons with the intervention data. The importance of GA documentation was emphasized during this training and did result in an increase in maternity register GA recordings [24].

Early data collection revealed GA quality and accuracy concerns, therefore PTBi-EA senior staff from both countries convened to agree on the CRCT 28-day follow-up eligibility criteria. These criteria were: all babies less than 2500g and babies between 2500g and 3000g if the GA is reported as less than 37 weeks. These criteria were chosen for ease of use, and because they included all low birth weight (LBW) babies, likely to be preterm using INTERGROWTH 21$^{st}$ Newborn Birth Weight Standards (IG21-NBWS) data as a reference, and babies between 2500g and 3000g only if they had a registered GA less than 37 weeks [26]. This would capture more late preterm babies and exclude the majority of large babies that are unlikely to be preterm. While growth-restricted term babies were also likely to be included in the cohort of babies less than 2500g, the distinction between preterm and growth-restricted term babies was not possible to make without early ultrasound dating.

### Study design

This nested analysis was a retrospective chart review evaluating the completeness, consistency, and plausibility of the GA data in the maternity registers during the baseline period (March 1, 2016 –September 30, 2016) of the PTBi-EA CRCT. Eligibility criteria included all live and fresh stillbirth babies born at the 23 study facilities during baseline, with recorded GA, birth weight, and sex, that were greater than 24 and less than 42 weeks GA, and greater than 500g

and less than 6000g birth weight in order to compare the data to the IG21-NBWS [15]. Macerated stillbirths were excluded to comply with standard preterm birth rate definitions where live birth is the denominator. Fresh stillbirths, however, were included to parallel the PTBi-EA parent study in which they were included to account for early neonatal deaths misclassified as fresh stillbirths and to assess the impact of the intrapartum intervention package on fresh stillbirth rates.

## Data collection

A team of data collectors conducted line-by-line extraction of the maternity register data from each of the 23 facilities. All births were included in the dataset and data were transcribed as they were written by health providers. The data were entered into an Open Data Kit (ODK) tool and uploaded to a server hosted at UCSF. The datasets were combined and cleaned using Structured Query Language (SQL) and analyzed using RStudio (Version 1.0.136).

The GA data recorded in the maternity registers came from various sources, dependent on the individual midwife and data availability. Some were transcribed from ANC booklets provided by the mothers when they presented for labor, others were calculated from maternal-reported LMP or measured from fundal height, while others appeared to be adjusted based on informal post-partum provider assessments. It was unclear when the information was recorded in the register and seemed to vary dependent on the midwife, with some filling the information in throughout the shift and others filling it in batches from the patient charts at the end of a shift. Few women received ultrasounds during their pregnancy as they could only be obtained at private facilities through out-of-pocket expenditure and they were rarely received for the purpose of GA dating.

## Data analysis

Data completeness was calculated as a proportion of all births where GA, birth weight, sex, and birth outcome were recorded (looked at as separate variables and a combined variable for births where all four variables were complete). The consistency evaluation was conducted only for Kenyan data as their maternity registers list both a recorded GA and a separate LMP date, and the Ugandan registers listed only a recorded GA. We used Naegele's rule to create a "calculated GA" variable from the LMP date and date of delivery [27]. The differences between the calculated and recorded GAs were compared and those with a difference of less than one week were considered to be equal. Descriptive statistics of calculated GA and the GA differences were included and a Bland-Altman plot displays the differences graphically.

Plausibility of GAs were evaluated by calculating the percentage of births where the birth weight for a given GA fell within the 3rd and 97th percentiles according to the IG21-NBWS data. Any birth that fell outside of these boundaries was considered to have an implausible GA. As GAs were recorded in whole weeks the 3rd percentile of week, 0 days and the 97th percentile of week, 6 days was used. For example, a female baby with a GA of 30 had a range from 900g (3rd percentile for 30 weeks, 0 days) to 2070g (97th percentile for 30 weeks, 6 days).

Finally, we calculated different approaches to estimating a preterm birth rate:

**Estimate #1: GA <37 weeks**–standard definition of preterm babies

**Estimate #2: GA <37 weeks with implausible GAs removed** (those above the 97th or below the 3rd IG21-NBWS birth weight percentiles for GA)–standard definition of preterm babies with implausible GAs removed

**Estimate #3: Birth weight <2500g** –standard definition of LBW babies

**Estimate #4: A birth weight<2500g or a birth weight between 2500g – 3000g if the GA is <37 weeks (PTBi-EA CRCT eligibility criteria)**–using IG21-NBWS as a guide, all LBW babies were included and those with a birth weight above 3000g were excluded. GA was only considered for babies with a birth weight in between these two boundaries [23, 26].

## Ethical considerations

The IRB committees of the University of California, San Francisco (Study ID# 16–19162), the Kenyan Medical Research Institute (Study ID# 0034/321), and Makerere University School of Public Health (Study ID# 189) reviewed and approved this study. All data were stored on encrypted computers and servers. The subset of data used in this analysis was de-identified prior to access.

## Results

In both countries, GA was the least recorded variable of those evaluated in the maternity register (93.4% in Kenya and 71.5% in Uganda) followed by birth weight (96.4% in Kenya and 86.7% in Uganda). Ineligible births, per our definition, totaled 5.7% in Kenya and 16.2% in Uganda. The analytic datasets excluded ineligible births and births with missing variables which resulted in 4762 births in Kenya (88.8% of total births) and 8935 in Uganda (64.5% of total births) (Table 1).

Using the analytic datasets described in Table 1, GA histograms plots show 38 weeks as the most frequent GA in both countries, followed by 40 and 36 weeks in Kenya and 39 and 37 weeks in Uganda (Fig 1). Birth weight histograms plots similarly show 3000g as the most common birth weight in both countries followed by 2800g and 3200g in Kenya and 3500g and 3200g in Uganda (Fig 1).

**Table 1. Completeness of preterm birth variables recorded in facility-based maternity registers.**

|  | Kenya | Uganda | Total |
|---|---|---|---|
|  | n (%) | n (%) | n (%) |
| Total N | 5360 (100.0) | 13859 (100.0) | 19219 (100.0) |
| **Data completeness** | | | |
| Gestational age | 5006 (93.4) | 9906 (71.5) | 14918 (77.6) |
| Birth weight | 5166 (96.4) | 12022 (86.7) | 17188 (89.4) |
| Sex | 5176 (96.6) | 12324 (88.9) | 17500 (91.1) |
| Neonatal outcome* | 5231 (97.6) | 12648 (91.3) | 17879 (93.0) |
| All variables complete | 4841 (90.3) | 9136 (65.9) | 13977 (72.7) |
| **Ineligible births** | | | |
| Gestational age <24 weeks | 15 (0.3) | 97 (0.7) | 112 (0.6) |
| Gestational age >42 weeks | 17 (0.3) | 22 (0.2) | 39 (0.2) |
| Birth weight <500g | 202 (3.8) | 1837 (13.3) | 2039 (10.6) |
| Birth weight >6000g | 1 (0.0) | 3 (0.0) | 4 (0.0) |
| Macerated stillbirths | 69 (1.3) | 287 (2.1) | 356 (1.9) |
| Total ineligible | 304 (5.7) | 2246 (16.2) | 2550 (13.3) |
| **Dataset** | | | |
| Analytic dataset** | 4762 (88.8) | 8935 (64.5) | 13697 (71.3) |

*If the baby was live birth or stillborn.

**Births with gestational age, birth weight, or sex missing were removed as well as ineligible births.

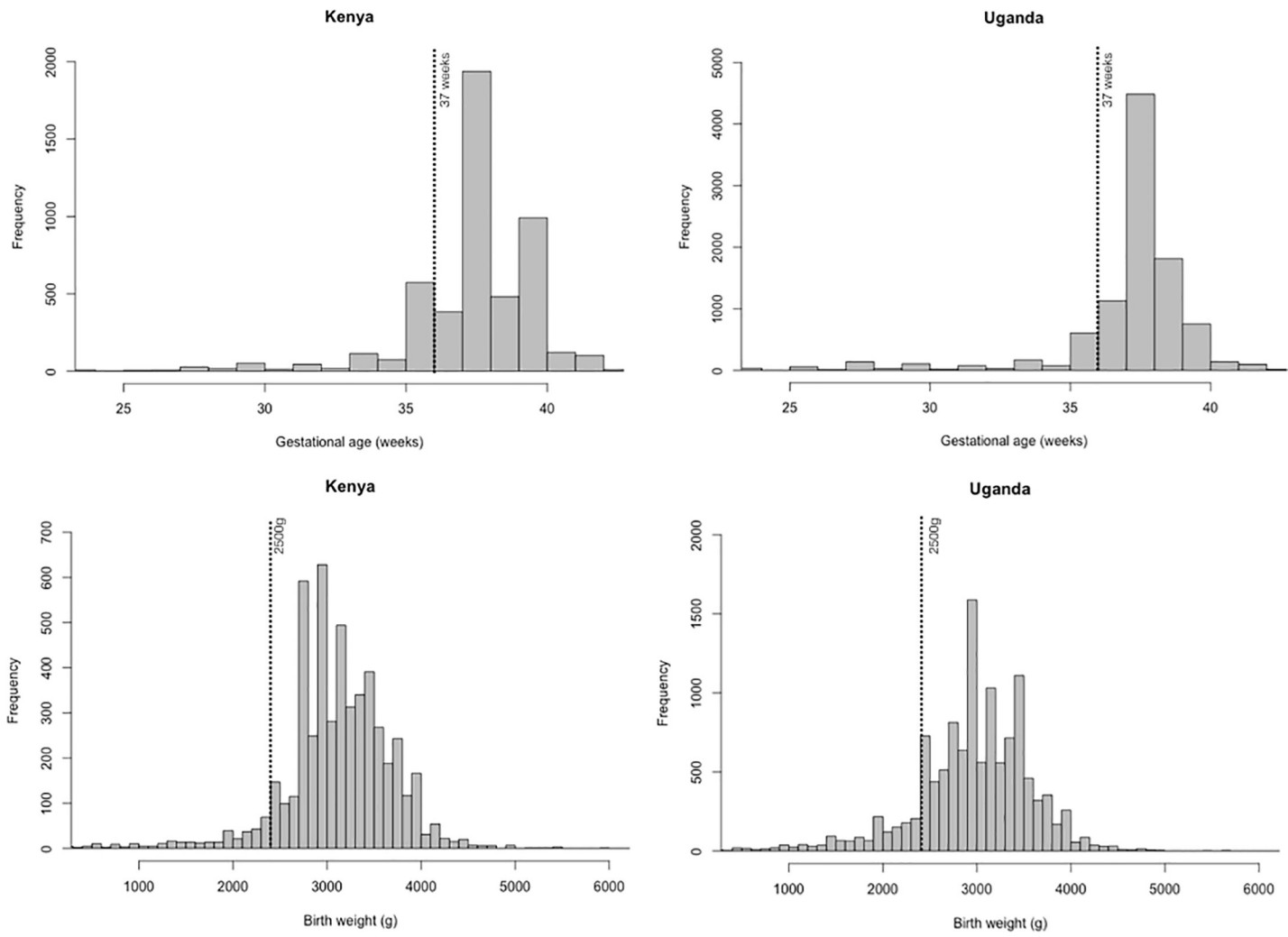

**Fig 1. Gestational age and birth weight histograms.**

The majority of maternity unit patients were between ages 18–25 years in both countries (53.7% and 54.9% in Kenya and Uganda, respectively). In Kenya, maternal age skewed younger with 13.4% of patients less than 18 years old compared to 6.6% in Uganda. Fresh stillbirths were 1.3% in Kenya and 2.8% in Uganda (Table 2).

Kenya's maternity registers recorded LMP in 85.5% of births. The calculated GAs ranged from -14.3 weeks to 91.9 weeks with a median of 39.1 weeks. Comparing the calculated GA to the recorded GA of the same birth, the differences ranged from 0.0 to 54.9 weeks with an average difference of 4.2 weeks. The recorded and calculated GAs were equal (with a difference less than one week) in 29.5% births (Table 3). The Bland-Altman plot shows the mean of the recorded and calculated GAs on the x-axis and the difference on the y-axis, displaying graphically the range of differences (Fig 2).

When comparing birth weight and GA data to the IG21-NBWS, 12.3% of births had implausible GAs (either below the 3rd or above the 97th birth weight percentile for GA) in Kenya and 11.4% in Uganda. In Kenya, more births were above the 97th percentile (9.2%) compared to below the 3rd percentile (3.1%). In Uganda, implausible GAs were evenly distributed above the 97th and below the 3rd percentiles (5.7% for both). In Kenya, births recorded as 27

**Table 2. Maternity unit patient demographics and outcomes as recorded in facility-based maternity registers.**

| | Kenya | Uganda | Total |
|---|---|---|---|
| | N (%) | N (%) | N (%) |
| Total N | 4762 | 8935 | 13697 |
| **Maternal age** | | | |
| <18 years | 638 (13.4) | 588 (6.6) | 1226 (9.0) |
| 18–25 years | 2559 (53.7) | 4905 (54.9) | 7464 (54.4) |
| 26–35 years | 1349 (28.3) | 2902 (32.5) | 4251 (31.0) |
| >35 years | 187 (3.9) | 501 (5.6) | 688 (5.0) |
| **Sex** | | | |
| Female | 2329 (48.9) | 4212 (47.1) | 6541 (47.8) |
| Male | 2433 (51.1) | 4723 (53.9) | 7156 (52.2) |
| **Outcomes** | | | |
| Alive | 4698 (98.7) | 8688 (97.2) | 13386 (97.7) |
| Fresh stillbirth | 64 (1.3) | 247 (2.8) | 311 (2.3) |
| **Mode of Delivery** | | | |
| Vaginal delivery | 4442 (93.3) | 6689 (74.9) | 11131 (81.3) |
| Cesarean section | 320 (6.7) | 2246 (25.1) | 2566 (18.7) |

weeks GA had the highest implausibility based on birth weight (85.7%), followed by 28 weeks (73.7%) and 31 weeks (72.7%). In Uganda, births recorded as 25 weeks were the most likely to be implausible based on birth weight (60.0%), followed by 28 and 31 weeks (both 56.3%) (Table 4). Fig 3 is a scatter plot of GA (x-axis) and birth weight (y-axis) data with the IG21-NBWS 3rd and 97th percentiles overlaid. The spread of birth weights for each GA as well as the relative proportions outside of the boundaries are displayed (Fig 3).

Preterm birth rates were calculated by various methods in Table 5. All GAs less than 37 weeks (estimate #1) resulted in a preterm birth rate of 18.5% in Kenya and 12.4% in Uganda. With the implausible GAs removed, the rates reduced to 13.1% in Kenya and 9.3% in Uganda (estimate #2). The LBW definition of <2500g with no GA data included resulted in rates of

**Table 3. Consistency\* of gestational age estimates recorded in facility-based maternity register.**

| | Kenya |
|---|---|
| | N (%) |
| Total | 4762 (100.0) |
| **Calculated gestational age+** | |
| Recorded last menstrual period | 4073 (85.5) |
| Range | -14.3 weeks, 91.9 weeks |
| Median GA | 39.1 weeks |
| Interquartile range | 4.6 weeks |
| **Difference between recorded and calculated gestational ages** | |
| Matching GAs\*\* | 1407 (29.5) |
| Median difference | 1.7 weeks |
| Interquartile range | 3.9 weeks |

\*Evaluated by comparing the gestational age recorded in the maternity register to a gestational age calculated based on last menstrual period

+Calculated by the formula: last menstrual period subtracted from date of delivery divided by seven.

\*\*Recorded gestational age and calculated gestational age for the same birth difference less than one week

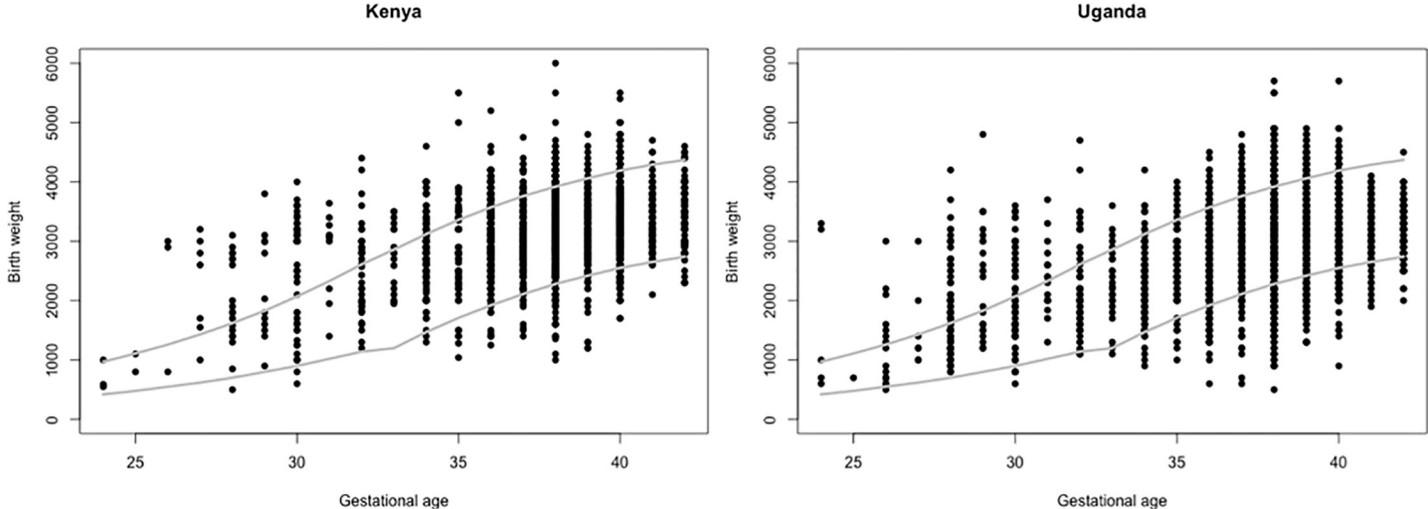

**Fig 2. Gestational age and birth weight scatter plots with INTERGROWTH-21$^{st}$ lines (3$^{rd}$ and 97$^{th}$ birth weight percentiles for GA).**

6.5% in Kenya and 11.3% in Uganda (estimate #3). The PTBi-EA algorithm of all births less than 2500g and between 2500g and 3000g if the GA is less than 37 weeks (estimate #4) resulted in preterm birth rates of 11.5% and 14.4% (Table 5).

**Table 4. Plausibility\* of gestational ages recorded in facility-based maternity registers.**

| | Kenya | | | | Uganda | | | | Total | | | |
|---|---|---|---|---|---|---|---|---|---|---|---|---|
| GA | N | <3$^{rd}$ | >97$^{th}$ | Implausible GAs | N | <3$^{rd}$ | >97$^{th}$ | Implausible GAs | N | <3$^{rd}$ | >97$^{th}$ | Implausible GAs |
| 24 | 3 | 0 (0.0) | 1 (33.3) | 1 (33.3) | 5 | 0 (0.0) | 3 (60.0) | 3 (60.0) | 8 | 0 (0.0) | 4 (50.0) | 4 (50.0) |
| 25 | 2 | 0 (0.0) | 0 (0.0) | 0 (0.0) | 1 | 0 (0.0) | 0 (0.0) | 0 (0.0) | 3 | 0 (0.0) | 0 (0.0) | 0 (0.0) |
| 26 | 3 | 0 (0.0) | 2 (66.7) | 2 (66.7) | 17 | 1 (5.9) | 8 (47.1) | 9 (52.9) | 20 | 1 (5.0) | 10 (50.0) | 11 (55.0) |
| 27 | 7 | 0 (0.0) | 6 (85.7) | 6 (85.7) | 7 | 0 (0.0) | 2 (28.6) | 2 (28.6) | 14 | 0 (0.0) | 8 (57.1) | 8 (57.1) |
| 28 | 19 | 1 (5.3) | 13 (68.4) | 14 (73.7) | 96 | 0 (0.0) | 54 (56.3) | 54 (56.3) | 115 | 1 (0.9) | 67 (58.3) | 68 (59.1) |
| 29 | 12 | 0 (0.0) | 6 (50.0) | 6 (50.0) | 25 | 0 (0.0) | 14 (56.0) | 14 (56.0) | 37 | 0 (0.0) | 20 (54.1) | 20 (54.1) |
| 30 | 48 | 1 (2.1) | 32 (66.7) | 33 (68.8) | 87 | 5 (5.7) | 34 (39.1) | 39 (44.8) | 135 | 6 (4.4) | 66 (48.9) | 72 (53.3) |
| 31 | 11 | 0 (0.0) | 8 (72.7) | 8 (72.7) | 16 | 0 (0.0) | 9 (56.3) | 9 (56.3) | 27 | 0 (0.0) | 17 (63.0) | 17 (63.0) |
| 32 | 38 | 1 (2.6) | 19 (50.0) | 20 (52.6) | 67 | 4 (6.0) | 17 (25.4) | 21 (31.3) | 105 | 5 (4.8) | 36 (34.3) | 41 (39.0) |
| 33 | 17 | 0 (0.0) | 7 (41.2) | 7 (41.2) | 24 | 1 (4.2) | 5 (20.8) | 6 (25.0) | 41 | 1 (2.4) | 12 (29.3) | 13 (31.7) |
| 34 | 106 | 2 (1.9) | 34 (32.1) | 36 (34.0) | 144 | 6 (4.2) | 11 (7.6) | 17 (11.8) | 250 | 8 (3.2) | 45 (18.0) | 53 (21.2) |
| 35 | 71 | 6 (8.5) | 21 (29.6) | 27 (38.0) | 70 | 8 (11.4) | 6 (8.6) | 14 (20.0) | 141 | 14 (9.9) | 27 (19.1) | 41 (29.1) |
| 36 | 546 | 8 (1.5) | 92 (16.8) | 100 (18.3) | 545 | 45 (8.3) | 36 (6.6) | 81 (14.9) | 1091 | 53 (4.9) | 128 (11.7) | 181 (16.6) |
| 37 | 367 | 10 (2.7) | 38 (10.4) | 48 (13.1) | 1049 | 32 (3.1) | 59 (5.6) | 91 (8.7) | 1416 | 42 (3.0) | 97 (6.9) | 139 (9.8) |
| 38 | 1868 | 32 (1.7) | 109 (5.8) | 141 (7.5) | 4167 | 180 (4.3) | 179 (4.3) | 359 (8.6) | 6035 | 212 (3.5) | 288 (4.8) | 500 (8.3) |
| 39 | 465 | 19 (4.1) | 10 (2.2) | 29 (6.2) | 1687 | 86 (5.1) | 56 (3.3) | 142 (8.4) | 2152 | 105 (4.9) | 66 (3.1) | 171 (7.9) |
| 40 | 963 | 50 (5.2) | 34 (3.5) | 84 (8.7) | 703 | 111 (15.8) | 18 (2.6) | 129 (18.3) | 1666 | 161 (9.7) | 52 (3.1) | 213 (12.8) |
| 41 | 118 | 5 (4.2) | 3 (2.5) | 8 (6.8) | 133 | 12 (9.0) | 0 (0.0) | 12 (9.0) | 251 | 17 (6.8) | 3 (1.2) | 20 (8.0) |
| 42 | 98 | 12 (12.2) | 4 (4.1) | 16 (16.3) | 92 | 19 (20.7) | 1 (1.1) | 20 (21.7) | 190 | 31 (16.3) | 5 (2.6) | 36 (18.9) |
| **Total** | **4762** | **147 (3.1)** | **439 (9.2)** | **586 (12.3)** | **8935** | **510 (5.7)** | **512 (5.7)** | **1022 (11.4)** | **13697** | **657 (4.8)** | **951 (6.9)** | **1608 (11.7)** |

\*Using the INTERGROWTH-21$^{st}$ newborn birth weight percentiles for gestational age. An implausible GA is one in which the birth weight fell either below the 3$^{rd}$ percentile or above the 97$^{th}$ percentile for gestational age. Abbreviations: GA–gestational age.

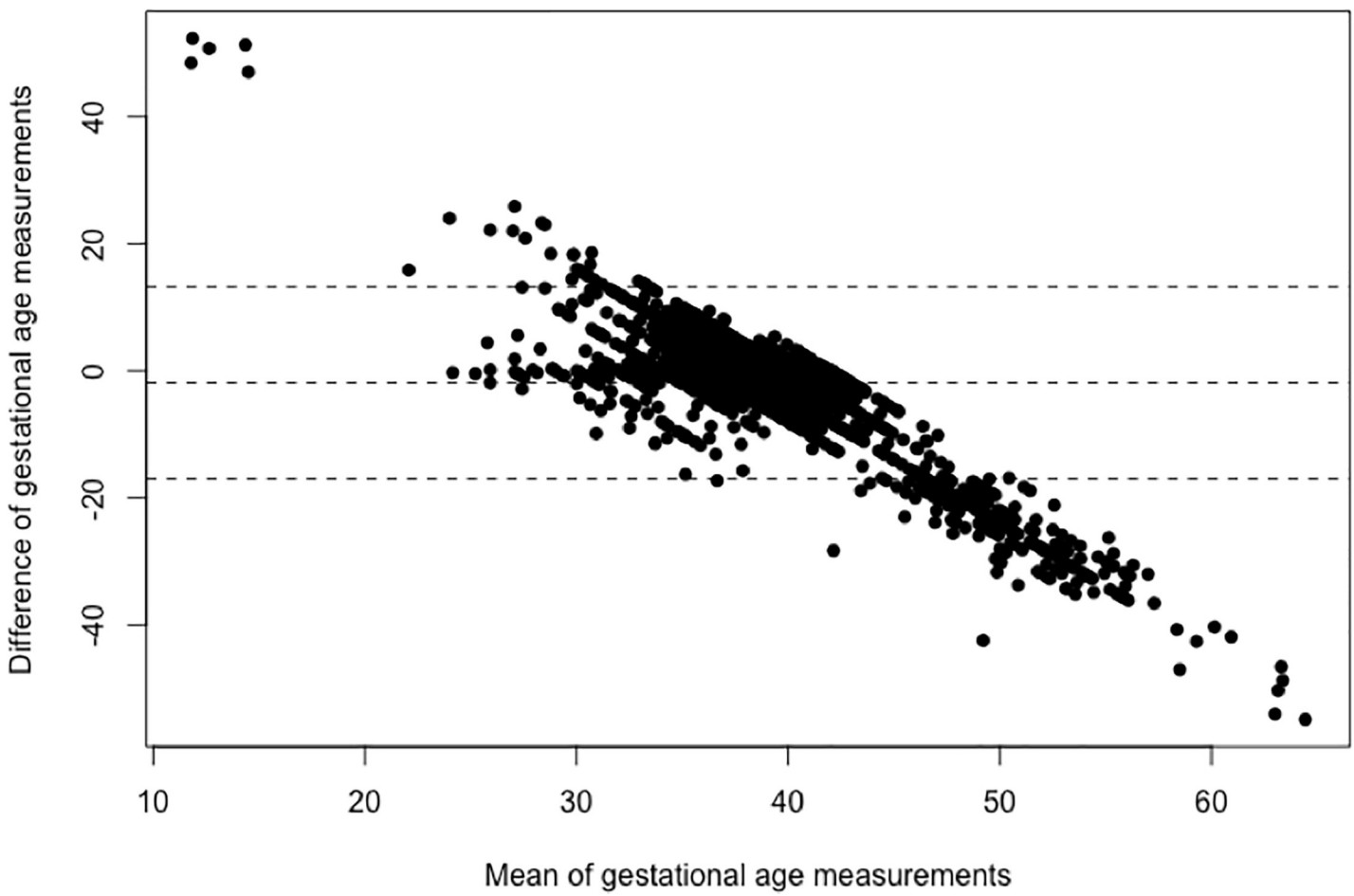

**Fig 3. Bland-Altman plot of Kenyan recorded and last menstrual period-calculated gestational ages.**

## Discussion

Lacking a gold standard, PTBi-EA sought an alternative method to assess preterm birth eligibility criteria using facility routine data sources. The PTBi-EA inclusion criteria (estimate #4)

**Table 5. Preterm birth rate estimates using various gestational age and birth weight criteria.**

| | Kenya | Uganda | Total |
|---|---|---|---|
| | n (%) | n (%) | n (%) |
| **Total N** | **4762 (100.0)** | **8935 (100.0)** | **13697 (100.0)** |
| **Estimate #1** | 883 (18.5) | 1104 (12.4) | 1987 (14.5) |
| (<37 weeks GA) | | | |
| **Estimate #2** | 623 (13.1) | 835 (9.3) | 1458 (10.6) |
| (<37 weeks GA, inaccurate GAs removed) | | | |
| **Estimate #3** | 308 (6.5) | 1011 (11.3) | 1319 (9.6) |
| (<2500g) | | | |
| **Estimate #4** | 550 (11.5) | 1291 (14.4) | 1841 (13.4) |
| (<2500g & 2500g – 3000g if GA is <37 weeks)* | | | |

*East Africa Preterm Birth Initiative study eligibility criteria.

of all babies less than 2500g and babies between 2500g and 3000g if the GA listed is less than 37 weeks, was an imperfect solution to a complicated problem, but one that was cost efficient and practical. It was simple to calculate by the PTBi-EA study nurses and yielded reasonable baseline preterm birth rates in both countries per existing estimates. While the PTBi-EA (estimate #4) preterm birth rates of 11.5% in Kenya and 14.4% in Uganda are substantially higher than the recent national estimates of 8.6% and 6.6% in Kenya and Uganda, respectively, by Chawanpaiboon/Vogel (2019), this is not surprising given the location and patient population of the PTBi-EA study sites [28]. Chawanpaiboon/Vogel (2019) acknowledge the limitations in their estimates given that both countries have little or no available national civil registration and vital statistics for preterm birth and therefore study data were used to build their statistical models, generally from research conducted in the capital cities [28]. Both Migori County and the Busoga region of Uganda are rural areas where mothers have fewer lifetime interactions with the healthcare system, are more likely to be malnourished and anemic, receive fewer antenatal care visits, and are more likely to be exposed to infection, particularly malaria, than women in the capital cities of Nairobi and Kampala, all of these factors increasing the risk of preterm birth [7, 29]. Few other published national estimates exist due to the paucity of accurate GA data, but the Blencowe (2012) national estimates from 2010 were 12.3% in Kenya and 13.1% in Uganda and the IG21 (2015) Kenya (with data from hospitals in Parklands, a wealthy suburb of Nairobi) data showed an 8.3% preterm birth rate [30,31]. It should be noted that our inclusion of fresh stillbirths veers from the traditional preterm birth definition, as recommended by others, and this may contribute to further differences between these estimates [17, 32].

In this paper, we also quantified maternity register GA data quality issues through completeness (Table 1), consistency (Table 3), and plausibility (Table 4) evaluations and calculated preterm birth rate estimates using various methods. In the PTBi-EA dataset, GA was the most incomplete variable in both countries. Recorded GA was only consistent with calculated GA in 29.5% of births in Kenya, meaning the remaining births were either not calculated from LMP, had an inaccurate LMP, or the GA was calculated incorrectly. At least 11.7% of births in the two countries combined had implausible GAs with potentially more given that falling within the IG-NBWS boundaries meant only that the GA was plausible, not necessarily accurate.

Given the data quality issues described for GA, other variables were considered to increase the accuracy of identifying preterm neonates, and thus preterm birth rates. Without a gold standard we cannot calculate the sensitivity and specificity of these estimates, but we can evaluate each one for its likely accuracy and practicality of use in an implementation science study. In Kenya, estimate #1 (<37 weeks recorded GA) resulted in a preterm birth rate of 18.5%. Considering the high number of births above the 97th percentile, it appears that health workers are underestimating GA and therefore overestimating preterm birth. Whether this is intentional or not is unknown but seems to most likely be the result of inaccurate LMP data leading to an inaccurate GAs. Uganda's estimate #1 of 12.4% is less straight forward given the balance of implausible GAs above and below the cut-offs, however with a LBW rate of 11.3% in Uganda, it seems possible that this is an underestimation, missing some of the older, heavier preterm babies.

Removing GAs with a birth weight above or below the IG-NBWS cut off and then using the less than 37 weeks GA (estimate #2) is a reasonable approach and results in preterm birth estimates of 13.1% in Kenya and 9.1% in Uganda. This approach, however, would have been impractical to implement as part of study protocol as it would have required detailed algorithms and made eligibility determination complicated for the study nurses with competing clinical duties. As such, this may be a more applicable approach to surveillance estimates.

According to the IG21-NBWS, the majority of babies less than 2500g are also likely to be preterm [26, 33]. As such, using the LBW definition (estimate #3) of all babies less than 2500g

is a natural proxy for preterm birth when GA data are poor for the purposes of post-partum clinical interventions, surveillance, or participation in trials. It is a simple assessment easily conducted by a busy health worker and uses what may be the more reliable variable of birth weight. It does, however, miss late preterm infants that are not LBW babies who are still at risk of prematurity related adverse outcomes, particularly in low-resource settings, and also subsequently includes some growth-restricted term babies [34].

Estimate #4, combining GA and birth weight, includes all LBW babies and uses 3000g as a maximum cut-off. GA was only considered in babies with a birth weight between 2500g and 3000g which led to a higher probability of accuracy given the birth weight restrictions. This was a straight-forward algorithm that made determining eligibility simple and less time consuming.

Despite the utility of the proposed approaches to identify preterm birth, future interventions are needed to improve GA accuracy in Migori County and the Busoga region. These might include earlier engagement in ANC, GA dating at first ANC visit, better systems of capturing ANC data such that the intrapartum provider can access the data without relying on the patient to her bring her own records, and more in-depth training for health care workers on the IG21-NBWS data to allow for GA adjustments based on birth weight (ie., if the LMP calculates to a 28-week GA but the baby has a birth weight of 4,000g, going back to the mother to probe for a more accurate LMP). Intrapartum calculations of GA are likely to be poor whether they are done from fundal height, LMP, or even ultrasound when compared to the same calculations made in the first trimester. These interventions could greatly increase the accuracy of GA without the investment of ultrasound dating for all women.

## Limitations

The main methodological challenge of this study was the commitment to using routinely collected data sources rather than establishing a parallel research study data collection approach, but as an implementation science study it was central to the study as a whole to work with the data streams and systems already in place.

Differences in data quality were seen between the two countries, which likely had to do with the increased volume at the facilities in Uganda which were higher level hospitals as compared Kenya which included lower level health facilities as well as hospitals.

As GA could not be confirmed, it is not possible to calculate the percentage of small-and large-for-GA babies captured in the PTBi-EA estimates and there is a high likelihood that growth-restricted term babies were included in the <2500g category. In low-resource settings, however, this distinction rarely results in differential clinical care. Newborn units in PTBi-EA sites tend to admit babies based on either birth weight or clinical status, not GA, and lack the technological and human resources of a neonatal intensive care unit where clinical care would differ for a preterm versus growth-restricted term baby. Additionally, there were natural errors in data due to the hand recording in the maternity register by busy frontline health workers. GA was also recorded in whole weeks which limited the granularity of the IG21-NBWS analysis. Birth weights, although all using digital scales, were also measured in different facilities and scales may not have been routinely calibrated. Finally, the data strengthening activities that were launched during the baseline period likely affected the completion and possibly consistency and plausibility to a lesser extent of the GA and birth eight data. These effects over time were further explored in other PTBi-EA publications [24].

## Conclusion

In 1975, the World Health Organization (WHO) reclassified "prematurity" defined by birth weight (<2500g) to "preterm" defined by GA (<37 weeks) [1, 35]. This reflected new

understandings of the differences between fetal growth and fetal maturation, but limited the utility of the definition in resource-poor settings where GA is substantially more challenging to assess than birth weight. Gestational age assessments, therefore, continue to be challenging in low-resource settings with both antenatal and postnatal measurements flawed in accuracy and practical applicability. Lacking a current alternative, estimate #4 gave PTBi a reasonable assessment of preterm birth rates and a simple, cost-effective eligibility assessment. Identifying and counting preterm babies is a critical first step towards a better understanding of the complex syndrome of preterm birth and its pathologies, leading to a reduction in rates over time, and the saving of neonatal lives.

## Supporting information

**S1 Data.**
(XLSX)

## Acknowledgments

The authors would like to thank the data collection teams of the East Africa Preterm Birth Initiative and the maternity unit health workers of the 23 facilities in Migori, Kenya and the Busoga Region of Uganda. Additional methods assistance was provided by Dr. Nancy L. Sloan and the data were managed and cleaned by Ryan Keating and Rikita Merai.

## Author Contributions

**Conceptualization:** Lara Miller, Phillip Wanduru, Nicole Santos, Elizabeth Butrick, Peter Waiswa, Phelgona Otieno, Dilys Walker.

**Data curation:** Lara Miller.

**Formal analysis:** Lara Miller.

**Funding acquisition:** Dilys Walker.

**Investigation:** Lara Miller, Peter Waiswa, Phelgona Otieno, Dilys Walker.

**Methodology:** Lara Miller, Phillip Wanduru, Nicole Santos, Elizabeth Butrick, Peter Waiswa, Phelgona Otieno, Dilys Walker.

**Project administration:** Lara Miller, Elizabeth Butrick, Peter Waiswa, Dilys Walker.

**Software:** Lara Miller.

**Supervision:** Peter Waiswa, Phelgona Otieno.

**Visualization:** Lara Miller.

**Writing – original draft:** Lara Miller.

**Writing – review & editing:** Lara Miller, Phillip Wanduru, Nicole Santos, Elizabeth Butrick, Peter Waiswa, Phelgona Otieno, Dilys Walker.

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
