## [Decision Letter · Decision Letter 0]

11 Jun 2020

PONE-D-20-06884

Working with what you have: how the East Africa Preterm Birth Initiative adjusted to limited gestational age data in routinely collected birth registers

PLOS ONE

Dear Dr. Miller,

Thank you for submitting your manuscript to PLOS ONE. After careful consideration, we feel that it has merit but does not fully meet PLOS ONE’s publication criteria as it currently stands. Therefore, we invite you to submit a revised version of the manuscript that addresses the points raised during the review process.

The manuscript has been reviewed by three colleagues. All suggest more clarity in the methods and detail with respect to how and when the systems strengthening intervention was implemented. Please note that there are 2 attachments with reviewer comments.

We look forward to receiving your revised manuscript.

Kind regards,

Emma K. Kalk

Academic Editor

PLOS ONE

Journal Requirements:

3. Please note that in order to use the direct billing option the corresponding author must be affiliated with the chosen institute. Please either amend your manuscript or remove this option (via Edit Submission).

Reviewers' comments:

Reviewer's Responses to Questions

**Comments to the Author**

1. Is the manuscript technically sound, and do the data support the conclusions?

Reviewer #1: Yes

Reviewer #2: Yes

Reviewer #3: Yes

2. Has the statistical analysis been performed appropriately and rigorously? 

Reviewer #1: Yes

Reviewer #2: Yes

Reviewer #3: Yes

3. Have the authors made all data underlying the findings in their manuscript fully available?

Reviewer #1: Yes

Reviewer #2: Yes

Reviewer #3: Yes

4. Is the manuscript presented in an intelligible fashion and written in standard English?

Reviewer #1: Yes

Reviewer #2: Yes

Reviewer #3: Yes

5. Review Comments to the Author

Reviewer #1: Thank you for allowing me to review PONE-D-20-06884 “Working with what you have: how the East Africa Preterm Birth Initiative adjusted to limited gestational age data in routinely collected birth registers”. This well-written paper that evaluated methods for determining preterm birth at baseline before an RCT to address PTB in Uganda and Kenya. They chose a pragmatic approach (combination of birth weight and GA) in a large sample where fetal ultrasounds, the gold standard, are not routinely used.

Major:

1. Abstract, lines 54-7: “while reinforcing and strengthening facility, routine data sources,” change to “while reinforcing and strengthening facilities the East Africa Preterm Birth Initiative (PTBi-EA) evaluated the quality of routine data sources that combined GA and birth weight to assess preterm birth rates”. As written, this sentence seems to say you were strengthening data sources, which is not the aim of the study.

2. Introduction: clear and well-written. While I understand the point is to examine existing records, the US is the gold standard. It would be good to evaluate in the intro a bit more on the triangulation of methods between US/LMP/Newborn assessments like the Ballard. Which is the best for identifying those that are on the line between <37 and >37? Obviously easy to classify VLBW or very PT, but what about those a little below the cut-off points that lead to misclassification?

3. Do you have any data at all about ultrasound use in these facilities? Please provide to understand the absence of a gold standard. Also please provide brief info on newborn assessments, like the Ballard.

4. Lines 116-125: A bit concerned that this is baseline, but there seems to have been some kinds of intervention during baseline “The team conducted initial data strengthening in April (Uganda) and June (Kenya) 2016”. Did this influence data quality? Can you assess that, perhaps in a sensitivity analysis based on time before and after?

5. Lines 129-34 and Lines 258-263: What fraction of babies <2500 actually are PT rather than SGA, but term? You might look just at Kenya and Uganda, because in other countries, like India, have seen >40% LBW but only 9% PT by U/S.

6. Table 1: What are “outcomes”? Give examples. Are these outcomes different than those listed in Table 2?

7. Figure 1: Is there evidence of rounding up to 2500 grams to make outcomes appear more favorable (e.g less LBW)? Would be good to add a dotted line to histograms at 2500 grams

8. Recommend a Bland-Altman plot of agreement/differences between recorded and calculated Gas

9. Table 3: Add “recorded” in front of LMP; Is the difference normally distributed? Would it be better to use the median?

10. Line 232: Could you say which of your estimations illustrate completeness, consistency, and plausibility evaluations?

11. Line 248: “It appears that health workers are underestimating GA and therefore overestimating preterm birth.”-Do you know why that might be? Does that lead to more funding or more resources? I know that sometimes it benefits to “undercount” if you want data to look better, but why would they want to make data look “worse”?

12. Line 254-55: why would it be impractical to implement?

13. Line 265-7: Please compare #4 estimates to national reported data.

Minor

1. Title: “Adjusted to” seems strange, I think of statistical adjustment. I think this could be reworded

2. Line 85: late to care is another reason for poor dating.

3. Line 81: but not low or late prenatal care? Perhaps that is not a cause of PTB, but just causes problems with dating GA?

4. Line 139: add “live” before births, since that is the denominator (line 144).

5. Line 140: GA < 24 and > 42 weeks are not plausible? I think they are, especially < 24, which might become fresh stillbirths.

6. Line 152-52: Cite reference. I am not sure how dividing the difference in dates by 7 creates the calculated EDD. I think you mean the difference in total days of pregnancy (280 days being the perfect = between actual and estimates)

7. Line 167 and line 224: not the same as the abstract, where <2,500 was used. Here you use a lower bound.

8. Line 176: add citations, here states CITE IG and protocol paper

9. Line 188: ineligible births ADD “per our definitions”

Reviewer #2: Please see attached document.

Simple but valuable evaluation. Technically sound piece of work.

Simple descriptive analysis but appropriate, does not require statistical review.

Authors have made dataset publicly available

Reviewer #3: The authors describe the results of a retrospective analysis evaluating the quality of gestational age data in maternity registers from study facilities in Kenya and Uganda. Estimation of preterm birth burden at population level is known to be complicated by data quality and availability, as such, this manuscript addresses an important public health concern.

In this manuscript, the authors report concerns about the quality of gestational age data in maternity registers; and present practical inclusion criteria for identifying preterm infants using routinely collected data when there are questions about GA data quality. I have attached my comments

6. PLOS authors have the option to publish the peer review history of their article (what does this mean?). If published, this will include your full peer review and any attached files.

Reviewer #1: No

Reviewer #2: Yes: Amy L. Slogrove

Reviewer #3: No

---

## [Author Response · Author response to Decision Letter 0]

26 Jul 2020

Response to Reviewers

Dear Editor, 

Thank you for the through and thoughtful response of the reviewers to our manuscript and for the opportunity to submit this revised version. We have copied each reviewer’s comments below (bold) and our response are below in plain text. We have made track changes where indicated and responded to each comment individually. We have included both the track changed version of the manuscript as well as a clean copy. 

Please let us know if there is any further information we can provide. 

Sincerely, 

Lara E. Miller 

EDITOR COMMENTS

Thank you for the comment, the style requirements have been updated according to the PLOS ONE formatting sample. 

Thank you for pointing out this oversight. The data can be found on UCSF’s DYAD platform at “Miller, Lara (2020), East Africa Preterm Birth Initiative Birth Register Data (March 2016 - October 2016), UC San Francisco, Dataset, https://doi.org/10.7272/Q6833Q63.” This link has been added to the revised cover letter. 

3. Please note that in order to use the direct billing option the corresponding author must be affiliated with the chosen institute. Please either amend your manuscript or remove this option (via Edit Submission).

As grantees of the Bill & Melinda Gates Foundation, we have registered this manuscript with their platform Chronos and direct billing can be processed through their site. The information on the PLOSONE platform has been updated. 

The caption for each figure was added to the manuscript and the .tiff files for each figure have been uploaded through the platform. 

REVIEWER’S COMMENTS: COMMENTS TO THE AUTHOR 

Reviewer #1 

Thank you for allowing me to review PONE-D-20-06884 “Working with what you have: how the East Africa Preterm Birth Initiative adjusted to limited gestational age data in routinely collected birth registers”. This well-written paper that evaluated methods for determining preterm birth at baseline before an RCT to address PTB in Uganda and Kenya. They chose a pragmatic approach (combination of birth weight and GA) in a large sample where fetal ultrasounds, the gold standard, are not routinely used.

Major:

1. Abstract, lines 54-7: “while reinforcing and strengthening facility, routine data sources,” change to “while reinforcing and strengthening facilities the East Africa Preterm Birth Initiative (PTBi-EA) evaluated the quality of routine data sources that combined GA and birth weight to assess preterm birth rates”. As written, this sentence seems to say you were strengthening data sources, which is not the aim of the study.

a. Thank you for the comment and for pointing out the confusion around this point. The PTBi-EA study did include strengthening of data sources as part of the package of interventions. Data strengthening and the other components of the PTBi-EA intervention have been further elaborated upon in lines 191 – 197.

2. Introduction: clear and well-written. While I understand the point is to examine existing records, the US is the gold standard. It would be good to evaluate in the intro a bit more on the triangulation of methods between US/LMP/Newborn assessments like the Ballard. Which is the best for identifying those that are on the line between <37 and >37? Obviously easy to classify VLBW or very PT, but what about those a little below the cut-off points that lead to misclassification?

a. Per the suggestion, we have we have expanded the introduction to include further reflection on non-ultrasound methods of assessing GA most often used in low resource settings. We have also included more references that show the varying degrees of accuracy of these methods when conducted correctly and the difficulty in implementing uniform accuracy across all healthcare workers. This new section can be found in lines 90 – 136. 

3. Do you have any data at all about ultrasound use in these facilities? Please provide to understand the absence of a gold standard. Also please provide brief info on newborn assessments, like the Ballard.

a. We have added a paragraph to the Methods section to address GA dating in the PTBi-EA study facilities (lines 239 – 245) and specially the existing availability of ultrasound for GA dating (lines 243 – 245). 

4. Lines 116-125: A bit concerned that this is baseline, but there seems to have been some kinds of intervention during baseline “The team conducted initial data strengthening in April (Uganda) and June (Kenya) 2016”. Did this influence data quality? Can you assess that, perhaps in a sensitivity analysis based on time before and after?

a. Thank you for bringing this to our attention. Data strengthening did occur during the baseline period in an effort to improve the quality of the baseline data for a more accurate comparison to the intervention period data. We pointed this out specifically in lines 193 – 197 and reference Keating (2019), a paper from our study further elucidating the data strengthening training and its effect on completeness of data recording. Additionally, we added the fact that data strengthening occurred during baseline to our Limitations in lines 536 – 538. 

5. Lines 129-34 and Lines 258-263: What fraction of babies <2500 actually are PT rather than SGA, but term? You might look just at Kenya and Uganda, because in other countries, like India, have seen >40% LBW but only 9% PT by U/S.

a. The reviewer brings up a good point, but unfortunately due to the lack of early ultrasound data throughout the country as a whole there is very little information available on growth-restricted babies. To address this point, we have added caveats on the likely inclusion of SGA term babies in the <2500g cohort in lines 208 – 210, 506 – 507, and 525 – 532. 

6. Table 1: What are “outcomes”? Give examples. Are these outcomes different than those listed in Table 2?

a. Thank you for pointing this out. Outcomes, in this case, refers to whether the baby was live birth or stillborn and is the same Table 1 and Table 2. We have updated the labelling in Table 1 to better clarify this point. 

7. Figure 1: Is there evidence of rounding up to 2500 grams to make outcomes appear more favorable (e.g less LBW)? Would be good to add a dotted line to histograms at 2500 grams

a. There does seem to be some instances of rounding up to the 2500g mark, but the reasons for this are unknown. It could be through provider choice or due to the types of weighing scales used. Thank you for the suggestion of the dotted line for 2500g, we have added this to Figure 1 as well as dotted lines for 37 weeks GA. 

8. Recommend a Bland-Altman plot of agreement/differences between recorded and calculated Gas

a. Thank you for the suggestion, this figure was added as Figure 2. 

9. Table 3: Add “recorded” in front of LMP; Is the difference normally distributed? Would it be better to use the median? 

a. We have added “recoded” in front of LMP. Thank you for the suggestion of using median rather than mean. This has been updated and in interquartile range added for both the calculated GA and the differences between the two GA estimates. Table 3. 

10. Line 232: Could you say which of your estimations illustrate completeness, consistency, and plausibility evaluations?

a. We have imbedded the corresponding tables into this sentence. 

11. Line 248: “It appears that health workers are underestimating GA and therefore overestimating preterm birth.”-Do you know why that might be? Does that lead to more funding or more resources? I know that sometimes it benefits to “undercount” if you want data to look better, but why would they want to make data look “worse”?

a. The reviewer brings up an interesting point. It is unclear if this overestimation was intentional or not, but due to the quality of the LMP data and the tendency to use that data as the means to calculate GA in the Kenyan facilities, we believe it is most likely unintentional. We have added a sentence addressing this point in lines 486 – 487. 

12. Line 254-55: why would it be impractical to implement?

a. Using the estimate #2 approach would have required more complex algorithms that would have made determining eligibility more labor intensive for the study nurses who had competing clinical duties. We have added a sentence in lines 494 – 495 addressing this point. 

13. Line 265-7: Please compare #4 estimates to national reported data.

a. Due to the paucity of GA data, few national estimates exist for either country. We have expanded the comparison to the Vogel/Chawainpaboon (2019) estimates as well as added the Blencowe (2012) national estimates and the INTERGROWTH-21st estimate for Kenya in lines 451 – 461. 

Minor

1. Title: “Adjusted to” seems strange, I think of statistical adjustment. I think this could be reworded

a. We agree with this assessment and have changed the title to, “Working with what you have: how the East Africa Preterm Birth Initiative used gestational age data from facility maternity registers.” 

2. Line 85: late to care is another reason for poor dating.

a. Thanking for this suggestion. We have addressed late ANC attendance in lines 93 – 95, 106 – 107, and 535 – 545. 

3. Line 81: but not low or late prenatal care? Perhaps that is not a cause of PTB, but just causes problems with dating GA?

a. Thank you for pointing this out. We have added more information on ANC attendance as referenced above.

4. Line 139: add “live” before births, since that is the denominator (line 144).

a. “Live” was added before birth in this line. 

5. Line 140: GA < 24 and > 42 weeks are not plausible? I think they are, especially < 24, which might become fresh stillbirths.

a. Thank you for pointing this out. While these are plausible GAs, the INTERGROWTH-21st Newborn Birth Weight Standards only report birth weight data for GAs ranging from 24 – 42 weeks. As such, we only included these GAs align with the INTERGROWTH data for the analysis in Table 4. We listed these in Table 1 as ineligible and clarified our eligibility criteria in lines 214 – 222. 

6. Line 152-52: Cite reference. I am not sure how dividing the difference in dates by 7 creates the calculated EDD. I think you mean the difference in total days of pregnancy (280 days being the perfect = between actual and estimates)

a. We appreciate highlighting this confusion. Nagel’s rule was used to make these calculations. This has been added in lines 252 – 253 and the reference added. 

7. Line 167 and line 224: not the same as the abstract, where <2,500 was used. Here you use a lower bound.

a. Thank you for highlighting this typo. The section in the Methods was updated to reflect the correct description in the Abstract. 

8. Line 176: add citations, here states CITE IG and protocol paper

a. Thank you for pointing out this typo, the comment has been deleted and the references added. 

9. Line 188: ineligible births ADD “per our definitions”

a. We added “per our definitions” to this line. 

Reviewer #2: Please see attached document .

Simple but valuable evaluation. Technically sound piece of work.

Simple descriptive analysis but appropriate, does not require statistical review.

Authors have made dataset publicly available

Working with what you have: how the East Africa Preterm Birth Initiative adjusted to limited gestational age data in routinely collected birth registers

Date: 09 June 2020

Thank you for the opportunity to review this manuscript by Miller and colleagues. Generally, this is a well written paper that clearly articulates the challenges of accurate preterm birth ascertainment in low resourced settings without access to early antenatal ultrasound and evaluates a pragmatic approach to improving the accuracy of preterm birth outcome classification through an implementation science study. Although the approach is still imperfect, the manuscript will be helpful to others working with similar quality gestational age information to understand the limitations.

Major comments:

• Data strengthening occurred in April in Uganda and in June in Kenya, with data collection spanning this period from March-September 2016. 

o Was any impact seen on the quality of the data before and after the data strengthening in each country? This would be quite helpful to understand whether investment in training to improve accuracy is effective and worthwhile. 

Significant increase in the competition of GA data was seen as a result of the data strengthening intervention. This was chronicled in greater detail by the Keating (2019) and this reference was added in line 197.We also added this as a limitation as a pre-post evaluation was not included in this analysis. 

o Considering large inter-country variation in gestational age estimation accuracy and ineligible births, were there any differences in the format, content or timing of training in the 2 different countries that can be learnt from to improve GA accuracy.

The data strengthening training was identical in both settings, but the initial data was less complete in Uganda likely due to the larger volume of the facilities where competing clinical duties lead to less time available for data transcription. Improvements were seen over time, however, as documented in Keating (2019), likely as a result ongoing reinforcement through the QI collaborative, the PRONTO simulation trainings, as well as ongoing data strengthening support. 

• Line 131 – To consider that term infants with fetal growth restriction could also be <2500g at birth. This might be particularly relevant in countries with high prevalence of smoking and alcohol in pregnancy that contribute to fetal growth restriction. In these countries preterm birth could be over-estimated using the PTBi-EA criteria.

o Thank you for highlighting this point. We have added caveats on the likely inclusion of SGA term babies in the <2500g cohort in lines 208 – 210, 506 – 507, and 525 – 532. 

• Regarding ineligible births (reflected in Table 1) - Are there differences by country in considering <500g as miscarriages and not recording in them in the maternity register? If a country more often includes miscarriages in the maternity register, then this could account for the higher proportion of “ineligible births”.

o This is an important point to consider. Technically, both countries count babies less than 1,000g as miscarriages (even if the babies survive). If a woman presents and is clearly having a miscarriage, she may be referred to the Gynecology ward rather than the Maternity Ward and recorded in that register. This practice may be happening less often in Uganda leading to the higher rates of babies <500g than in Kenya, but this is unclear. 

Minor comments and corrections:

• Abstract Objective 2nd sentence – think it should reach “chose an eligibility criterion (singular) that…” or otherwise drop the an “chose eligibility criteria that…”

o Thank you for bringing this to our attention. This change has been made. 

• Line 110 – might be clearer to the reader to specify that the intrapartum and immediate postnatal quality of care improvement package was specifically for women in preterm labour (if this is correct)

o We have added a section further explaining the PTBi-EA package of intervention in the Methods section in lines 191 – 197. 

• Line 176 – include citations 

o The indicated references have been added and thank you for pointing out this typo. 

Reviewer #3 

The authors describe the results of a retrospective analysis evaluating the quality of gestational age data in maternity registers from study facilities in Kenya and Uganda. Estimation of preterm birth burden at population level is known to be complicated by data quality and availability, as such, this manuscript addresses an important public health concern.

Comments to Authors

The authors describe the results of a retrospective analysis evaluating the quality of gestational age data in maternity registers from study facilities in Kenya and Uganda. Estimation of preterm birth burden at population level is known to be complicated by data quality and availability, as such, this manuscript addresses an important public health concern.

In this manuscript, the authors report concerns about the quality of gestational age data in maternity registers; and present practical inclusion criteria for identifying preterm infants using routinely collected data when there are questions about GA data quality. 

I have the following major and minor comments:

Introduction 

1. This section could perhaps be strengthened by highlighting the following 

- central importance of gestation age and its correlation with developmental processes 

i. Thank you for this suggestion. We have added more information on GA estimation methods in the Introduction and included information on developmental processes in the Discussion section, in lines 563 – 567. 

- inherent imprecision of gestational given its control by multiple physiological processes (natural variability and/or estimation error)

i. We have added further information on various non-ultrasound GA estimations in lines 96 – 136 and the imprecision in all methods for various reasons including physiological processes, specifically addressing variations in menstrual cycles in lines 103 – 104. 

Methods

1. “Methods and procedures are described in detail elsewhere”

- While the authors direct readers to the published study protocol for further detail, there are details of the procedures of the primary study that are central to this analysis that could be included or presented clearer to contextualize this analysis e.g. Intervention delivery at facility level and outcome assessment at facility and individual level could be useful information as well as the primary outcome of the primary study 

• Thank you for the comment. We have added a more detailed explanation of the parent study methods and the package of intervention in lines 180 – 210. 

2. Could the authors provide more information about the maternity registers 

- Are these completed at delivery? If completed at delivery what is the source of the data – is this based on self-report or patient-held records completed throughout the antenatal period? 

• Thank you for highlighting these important points. Further information about the maternity register was added in lines 244 – 246. 

• gestational age data: is this calculated by the healthcare workers, based on gestational age assessment completed at the first antenatal visit? Or is it based on an assessment done when women present at the facilities during early stages of labour? is there any indication whether GA recorded is based on antenatal or postnatal assessment?

a. The GA recorded in the register appeared to come from various sources dependent on the midwife and data availability. This is further explained in lines 240 – 244. 

• Is there any information of timing of presentation for antenatal booking? JT Price et al showed the importance of this information through their observation that bias in LMP based GA estimates increased as the gestation at antenatal care presentation advanced.

a. Although this information was not available in the maternity register, we added information on to the Introduction section on the number of women at the national level who seek ANC in the first trimmest and the average GA of first ANC visit (lines 93 – 95). 

3. “initial data strengthening in April (Uganda) and June (Kenya) 2016 which included a review of GA assessment methods….” and “….extraction of the maternity register data from each of the 23 facilities from March - September 2016”

- Could the authors indicate that this activity was part of the intervention packages of the primary study in order to “strengthen existing data collection processes in health facilities, introducing standard tools to improve GA assessment..”

• We added additional language on the timing of data strengthening component and how it was conducted during baseline in order to improve the quality of the baseline data for better comparison to the intervention (lines 193 – 197). 

- Since this data strengthening took place during data collection (particularly in Kenya) was the data completeness, consistency and plausibility – assessed before and after intervention ?

• This analysis looked only at the baseline period as a unified dataset and did not do a pre-post analysis. Keating et al (2019), however included this analysis in their evaluation of the impact of data strengthening activities. This reference was added to the Methods section in line 197. 

- Given the differences in timing of intervention – if differences were seen before and after the intervention perhaps this could be touched upon in the limitation section 

• Thank you for this suggestion. We have added this to the limitation section. 

4. “Completeness was calculated as a proportion of all births where GA, birth weight, sex, and birth outcomes were recorded” 

- Could the authors clarify that completeness was assessed for each of these variables separately

• Thank you for bringing this to our attention. We have clarified this point in lines 251 – 252.

5. “consistency evaluation was conducted only for Kenyan data as their maternity registers list both GA and LMP”

- Presumably the LMP is the date? Could the authors make this clearer. 

• Yes, the LMP is the date of the last menstrual period. This was clarified in line 253. 

- Is it known what the ‘recorded GA’ was based on? Is there information on method of assessment use?

• This seemed to vary per midwife and is further explained in lines 198 – 210. 

- Was there any GA assessment based on symphysis fundal height?

• It was unclear from the register data how each individual GA was assessed, but some did likely include fundal height assessments. This was further explained as part of the paragraph in lines 198 – 210. 

Results

1. Could the authors consider describing the ineligible births further in the results section 

- which exclusion criteria had the highest proportion of ineligible births? Were there any differences noted by country?

i. Births with a birth weight <500g accounted for the largest proportion of ineligible births in both countries but had a significantly higher rate in Uganda. It is unclear why this is the case but may be due to certain facilities having protocols for women experiencing miscarriage to be admitted to the Gynecology ward rather than the Maternity ward, but this was not explicitly clear. 

2. “The calculated GAs ranged from -14.3 to 91.9 weeks…”

- Were any of these LMPs excluded based on implausibility or were all included regardless of plausibility?

i. In order to display the quality concerns in the LMP data, we included all of the recorded LMPs even if they resulted in clearly implausible GAs. 

3. Is there any data on mode of delivery? – to enable distinction between provider initiated and spontaneous preterm births

i. Thank you for this comment. We have added mode of delivery to Table 2. 

4. “initial data strengthening in April (Uganda) and June (Kenya) 2016 which included a review of GA assessment methods….” 

- Were there improvements noted after the data strengthening intervention?

i. As stated above, we did not include a pre-post analysis of this invention but this analysis was conducted in the Keating (2019) paper. 

Discussion 

1. Highlighting the background rates of preterm births in Kenya and Uganda earlier in this section before the different inclusion criteria approaches are being discussed would be useful 

- Besides estimates presented by Chawanpaiboon et al are there any other preterm birth estimates reported from national statistics or from studies in the countries that could also be discussed in relation to estimates observed in this study?

Thank you for this suggestion. We have reordered the paragraphs and brought the comparison to national estimates to the beginning of the Discussion sections. 

2. Could the authors consider highlighting the following points

- the difficulty of preterm birth prevention given that it is a complex syndrome with interrelated aetiological factors and pathways contributing to phenotypical differences, and thus requiring a suite of evidence-based interventions

Thank you for this comment. This point has been added to our Conclusion in lines 562 – 572. 

- gestation based definitions of preterm birth are the most commonly used as they are better short- and long-term outcome predictors than measures of intrauterine growth. This could be followed by a discussion of the reasons why using gestation-based definitions is not always possible and the other methods available 

Further discussion of growth-restricted babies has been added to the Methods and Discussion, particularly in the Limitations section lines 547 – 554. 

3. “LBW definition…is a natural proxy for preterm birth when GA data are poor”

- Important to highlight that using LBW is a postnatal proxy which is useful for identifying preterm births in particular instances such as for clinical management as well as screening for additional postnatal clinical surveillance, interventions or trials such as the PTBi-EA.

Thank you for highlighting this point. We have added the sentence “As such, using the LBW definition (estimate #3) of all babies less than 2500g is a natural proxy for preterm birth when GA data are poor for the purposes of post-partum clinical interventions, surveillance, or participation in trials,” to lines 516 – 517. 

- While the authors mention that using birthweight may “miss late preterm infants that are not LBW” – perhaps a stronger point could be made that in instances where the goal is to accurately determine preterm birth this proxy measure maybe less useful because while approximately 2/3 of LBW infants are due to preterm birth, a proportion are term infants who are growth restricted 

To address this point, we have added caveats on the likely inclusion of SGA term babies in the <2500g cohort in lines 208 – 210, 506 – 507, and 525 – 532. 

4. Perhaps the authors could expand on “this pragmatic approach to screening allows research to be conducted without substantial investment of early pregnancy ultrasound” and underscore in the conclusion the need to invest in improving GA assessment methods during antenatal care and improving data quality of routinely collected data at population levels in order to improve identification of preterm infants

- We appreciate this comment and have added a section to the Discussion section highlighting the needed for earlier engagement in ANC and better ANC documentation that is more easily accessible to intrapartum providers, as all GA assessment methods are improved the earlier in pregnancy they are assessed (line 534 – 544). 

5. “…this is not surprising given that the PTBi-EA study sites were government hospitals In rural, malaria endemic areas”

- Could the authors expand on this point and discuss the discrepancy with Chawanpaiboon et al rates in more detail 

We have expanded our discussion around the comparison to Chwanpaiboon as well as other national estimates and how these national estimates (the few that exist) tend to use data from the capital cities whereas in the PTBi-EA sites in rural, malaria endemic areas, women experience more risk factors for preterm birth. 

6. “…in preterm birth estimates over 80% of stillbirths are also preterm ….”

- How were stillbirths classified? Were they all classified as preterm births

In our dataset, not all stillbirths had a GA and of those that did not all of the GAs were less than 37 weeks. Kramer (2012) and Blencowe (2013) both advocate for the inclusion of stillbirths in preterm birth numerators and denominators to capture this important cohort in preterm birth efforts. These references were added and the Discussion clarified in lines 458 – 460. 

7. Limitation section could be expanded to include a discussion of the methodological challenges of using routinely collected data for birth outcomes and how they relate specifically to PTBi-EA where enrolment occurs at delivery

- This section was expanded to discuss these suggested limitations and others related to growth-restricted babies and data strengthening activities. 

8. Interpretation of these findings could also be strengthen by 

- discussion of the potential effect of the data strengthening intervention – and whether it had any impact on the GA estimates or recording and how this exercise/intervention can be used to strengthen future studies using routinely collected data

A discussion of the data strengthening activities was added the limitations section in lines 558 – 560. 

- discussion about potential reasons for the differences in data quality between the two countries such as reasons for Uganda’s completeness plausibility 

Further discussion of the differences between the two countries was added to the limitation section in lines 551 – 553. 

Minor comments 

1. Evaluation of the “validity of the PTBi-EA eligibility criteria” is mentioned in the abstract as part of the aims of this analysis but not in the introduction 

a. We have added this discussion of the validity of the PTBi-EA eligibility criteria to the Introduction section. Specifically in lines 147 – 149. 

2. Line 111 – “Methods and procedures are described in detail elsewhere.” – check the reference 

a. Thank you for highlighting this point, the reference has been updated. 

3. Line 176 “(CITE IG and protocol paper) – correct this 

a. The referenced has been added and the comment removed.

---

## [Editor Report · Decision Letter 1]

31 Jul 2020

Working with what you have: how the East Africa Preterm Birth Initiative used gestational age data from facility maternity registers

PONE-D-20-06884R1

Dear Dr. Miller,

We’re pleased to inform you that your manuscript has been judged scientifically suitable for publication and will be formally accepted for publication once it meets all outstanding technical requirements.

Kind regards,

Emma K. Kalk

Academic Editor

PLOS ONE
---

## [Editor Report · Acceptance letter]

19 Aug 2020

PONE-D-20-06884R1 

Working with what you have: how the East Africa Preterm Birth Initiative used gestational age data from facility maternity registers 

Dear Dr. Miller:

I'm pleased to inform you that your manuscript has been deemed suitable for publication in PLOS ONE. Congratulations! Your manuscript is now with our production department. 

Kind regards, 

on behalf of

Dr. Emma K. Kalk 

Academic Editor

PLOS ONE